Genome-wide identification of candidate aquaporins involved in water accumulation of pomegranate outer seed coat

Liu Jianjian 1 2
Qin Gaihua 2 3
Liu Chunyan 2 3
Liu Xiuli 4
Zhou Jie 4
Li Jiyu 2 3
Lu Bingxin 1
Zhao Jianrong zhaojr@ahstu.edu.cn 1
1 College of Resource and Environment, Anhui Science and Technology University , Fengyang , China
2 Institute of Horticultural Research (Key Laboratory of Genetic Improvement and Ecophysiology of Horticultural Crop, Anhui Province), Anhui Academy of Agricultural Sciences , Hefei , China
3 Key Laboratory of Fruit Quality and Developmental Biology, Anhui Academy of Agricultural Sciences , Hefei , China
4 State Key Laboratory for Managing Biotic and Chemical Threats to the Quality and Safety of Agro-products, Zhejiang Academy of Agricultural Sciences , Hangzhou , China
Scheibe Renate
Electronic publication date: 2021 Jul 15
Publication date: 2021
Volume: 9
Electronic Location ID: e11810
Received 2020 Dec 30; Accepted 2021 Jun 27
Copyright: ©2021 Liu et al.
Copyright year: 2021
Copyright holder: Liu et al.
License: This is an open access article distributed under the terms of the Creative Commons Attribution License, which permits unrestricted use, distribution, reproduction and adaptation in any medium and for any purpose provided that it is properly attributed. For attribution, the original author(s), title, publication source (PeerJ) and either DOI or URL of the article must be cited.
License URL: https://creativecommons.org/licenses/by/4.0/

Keywords: Aquaporin, Phylogenetics, Water accumulation, Outer seed coat, Pomegranate

Funding: National Natural Science Foundation of China 32002123 Special project on Science and Technology of Anhui Province, China 201903b06020017 Major Science and Technology Projects in Anhui Province 18030701214 Biotechnology in Plant Protection of Ministry of Agriculture and Rural Affairs and Zhejiang Province This work was supported by grants from the National Natural Science Foundation of China (32002123), the Special project on Science and Technology of Anhui Province, China (201903b06020017), and the Major Science and Technology Projects in Anhui Province (grant no. 18030701214), and was funded by the Biotechnology in Plant Protection of Ministry of Agriculture and Rural Affairs and Zhejiang Province. The funders had no role in study design, data collection and analysis, decision to publish, or preparation of the manuscript.

==============================
Aquaporins (AQPs) are a class of highly conserved integral membrane proteins that facilitate the uptake and transport of water and other small molecules across cell membranes. However, little is known about AQP genes in pomegranate (Punica granatum L.) and their potential role in water accumulation of the outer seed coat. We identified 38 PgrAQP genes in the pomegranate genome and divided them into five subfamilies based on a comparative analysis. Purifying selection played a role in the evolution of PgrAQP genes and a whole-genome duplication event in Myrtales may have contributed to the expansion of PgrTIP, PgrSIP, and PgrXIP genes. Transcriptome data analysis revealed that the PgrAQP genes exhibited different tissue-specific expression patterns. Among them, the transcript abundance of PgrPIPs were significantly higher than that of other subfamilies. The mRNA transcription levels of PgrPIP1.3, PgrPIP2.8, and PgrSIP1.2 showed a significant linear relationship with water accumulation in seed coats, indicating that PgrPIP1.3/PgrPIP2.8 located in the plasma membrane and PgrSIP1.2 proteins located on the tonoplast may be involved in water accumulation and contribute to the cell expansion of the outer seed coat, which then develops into juicy edible flesh. Overall, our results provided not only information on the characteristics and evolution of PgrAQPs, but also insights on the genetic improvement of outer seed coats.

Introduction

Pomegranate (Punica granatum L.) is an important economic fruit tree species due to its functional and nutraceutical properties, and it is widely consumed as a fruit, juice, wine, and medicine (Johanningsmeier & Harris, 2011; Patel et al., 2008). Pomegranate is native to Iran, the northern India side of the Himalayan Mountains, and is widely planted in Mediterranean-like climates around the world, including Tunisia, Turkey, Spain, Egypt, Iran, Morocco, the USA, China, India, Argentina, Israel, and South Africa (Qin et al., 2017). The size of the juicy outer seed coat determines the edible quality of the fruit, so it has become an important fruit characteristic. Notably, the morphological characteristics of the pomegranate seed showed a compressed inner seed coat and an expanded outer seed coat, making it an appealing model for studying development of seed coats (Luo et al., 2020; Niu et al., 2018; Qin et al., 2020).

The development of the seed coat is accompanied by the import of organic and inorganic nutrients, including sugars, organic acids, cellulose, and water in developing seeds (Qin et al., 2020; Uçar & Karagöz, 2009; Zarei et al., 2016; Zhou et al., 2007). The symplastic pathway is one of the main pathways by which water moves, which is mediated by integral membrane proteins called aquaporins (AQPs), a major intrinsic protein family (Adams & Wendel, 2005; Lian et al., 2004; Suga et al., 2002). AQPs have been shown to transport small molecules such as carbon dioxide, glycerol, ammonia, urea, hydrogen peroxide, and selenite (Ivanov et al., 2007; Yasui et al., 1999; Zwiazek et al., 2017).

The general AQP structure is highly conserved in plants, and it is predicted to consist of six transmembrane (TM) α-helices (H1 to H6) and two half-helices (Deshmukh et al., 2015; Lee et al., 2005; Tornroth-Horsefield et al., 2006). These transmembrane α-helices are linked by five short loops (Loops A to E), Loops B and E contain the signature sequence Asparagine-Proline-Alanine (NPA) motif, which has the primary function of forming water-selective channels (Chaumont et al., 2001; Gupta & Sankararamakrishnan, 2009; Wallace & Roberts, 2004). Another important secondary structure known as an aromatic/arginine (ar/R) selectivity filter is formed by four residues contributed by transmembrane helices H2/H5 and the loop LE (Azad et al., 2016; Deshmukh et al., 2015; Hove & Bhave, 2011; Tornroth-Horsefield et al., 2006). These two structures predominantly determine the specificity of solute transport and transport rate (Deshmukh et al., 2015; Lee et al., 2005; Tornroth-Horsefield et al., 2006). According to sequence similarity and protein subcellular localization, AQPs in higher plant can be classified into five distinct subfamilies: plasma membrane intrinsic proteins (PIPs), tonoplast intrinsic proteins (TIPs), nodulin 26-like intrinsic proteins (NIPs) (Pommerrenig, Diehn & Bienert, 2015), small basic intrinsic proteins (SIPs), and unrecognized (or X) intrinsic proteins (XIPs) (Chaumont et al., 2001; Danielson & Johanson, 2010; Kaldenhoff & Fischer, 2006; Khabudaev et al., 2014; Quigley et al., 2002).

The AQP gene family has been widely studied in numerous plant species, such as Arabidopsis thaliana (Quigley et al., 2002), Oryza sativa (Sakurai et al., 2005), Glycine max (Zhang et al., 2013), cotton (Li et al., 2019), Populus trichocarpa (Gupta & Sankararamakrishnan, 2009), and grape (Fouquet et al., 2008), by blasting whole genome sequences. Previous studies have demonstrated that AQPs in higher plants play important roles in various physiological and development processes, such as cell and tissue expansion, fiber development, flower pollination, and seed development (Azad et al., 2004; Eisenbarth & Weig, 2005; Gattolin, Sorieul & Frigerio, 2011; Soto et al., 2008; Soto et al., 2010); (Van der Willigen et al., 2006; Wudick et al., 2014). For instance, in French bean (Phaseolus vulgaris), PvPIPs played a role in the import of water and nutrients by phloem-mediated transport and water recycling in the xylem in developing seeds (Terashima & Ono, 2002). In rice, OsPIP1;1 and OsPIP1;3 functions as water channels. Over-expression of OsPIP1;1 could increase rice yield and seed germination. Similarly, the overexpression of OsPIP1;3 promoted the speed of seed germination under drought conditions (Liu et al., 2013; Liu et al., 2007). In Pisum sativum L., PsNIP1 showed high expression level in developing pea seed coats, and the overexpression of PsNIP1 increased the uptake of water and glycerol (Schuurmans et al., 2003; Zhou et al., 2007). Interestingly, the highly correlations between the expression of AtTIP3/AtTIP1 and seed germination stages (a rapid imbibition of desiccated tissues and embryo growth, respectively) was observed, which also provided insight into the influence of AQPs on the seed developmental process (Gattolin, Sorieul & Frigerio, 2011). However, little is known about the role of AQPs on seed coat development.

In this study, we identified 38 PgrAQP genes in the pomegranate genome, then conducted gene structure, phylogenetics, and evolutionary divergence analyses. The expression patterns of PgrAQP genes in different tissues and seed developmental stages were analyzed, and the potential function of PgrPIP genes in response to drought stress was also identified. Furthermore, the candidate genes contributing to the water accumulation in the seed coat were selected based on a correlation analysis of gene expression and water accumulation in seed coats. Our findings provided fundamental information about the gene structure, phylogenetics, and evolutionary divergence of PgrAQP. In addition, this study also provided useful information for further investigation of the molecular mechanism by which PgrAQP functions in seed coat development.

Materials and Methods

Identification of the PgrAQP genes in the pomegranate genome

The whole genome sequences of pomegranate were downloaded from NCBI Genome (https://www.ncbi.nlm.nih.gov/genome/?term=Punica+granatum+L). Predicted protein sequences were obtained using TBtools with the following sets: Sequence Toolkit: Batch translate CDS to protein (Chen et al., 2020). The amino acid sequences of the AQPs in Arabidopsis and Populus trichocarpa were employed as queries to blast searches against the whole-genome sequences in pomegranate using TBtools with a query over 50% and e-value less than 0.01 (Quigley et al., 2002). The candidate sequences were submitted to NCBI for EST blast searches. Finally, the AQP protein sequences of pomegranate were examined to verify the presence of the characteristic MIP and transmembrane helical domains using the SMART program (http://smart.embl-heidelberg.de/) and TMHMM (http://www.cbs.dtu.dk/services/TMHMM/) (Finn et al., 2014; Letunic, Khedkar & Bork, 2020). The information of PgrAQP gene family (protein length, molecular weight, and isoelectric point) were determined using Expasy (https://web.expasy.org/compute_pi/) (Artimo et al., 2012). The subcellular localization of the PgrAQP proteins was predicted by WoLF PSORT (https://www.genscript.com/psort/wolf_psort.html) Cell-PLoc 2.0 (http://www.csbio.sjtu.edu.cn/bioinf/Cell-PLoc-2/)

Sequence alignment of PgrAQP genes, phylogenetic analysis, and classification

The predicted plant AQP protein sequences were aligned using the ClustalW tool in MEGA7.0 (Kumar, Stecher & Tamura, 2016). Arabidopsis thaliana, grape (Vitis vinifera), Populus trichocarpa and eucalyptus (Eucalyptus grandis) AQP protein sequences were collected from NCBI (https://www.ncbi.nlm.nih.gov/). A phylogenic tree was constructed with MEGA7.0 using the neighbor-joining method and bootstrap parameter with 1,000 replicates. For this study, Pomegranate AQPs were named according to the sequence homology and phylogenetic relationships with Populus trichocarpa . According to the name of the best hit gene in Populus trichocarpa, the subfamily classification (PIP, NIP, SIP, TIP, and XIP) and corresponding names of AQPs are shown in Fig. 1. TBtools software was used to determine the localization of PgrAQP genes on pomegranate chromosomes.

Figure 1 Phylogenetic analysis of AQP proteins from pomegranate, grape, Arabidopsis, Populus trichocarpa and Eucalyptus.

The tree was generated by the neighbor-joining (NJ) method in MEGA 7. 0 with 1,000 bootstrap replicates. Different background colors indicate the different subfamilies of the AQP proteins.

Gene structure and conserved motif analysis of PgrAQPs

The conserved motifs in the proteins were identified using Multiple Expectation Maximization for Motif Elucidation (MEME v4.12.0, http://meme-suite.org/tools/meme) with the following parameters: maximum number of motifs, 10; width of optimum motif, ≥15 and ≤50 (Bailey et al., 2009). The gene exon–intron structures of AQPs were confirmed using the GSDS database by aligning the open reading frame (ORF) with their corresponding genomic sequences (Hu et al., 2015a). Sequences representing NPA motifs, ar/R filter, and Froger’s residue positions were manually identified based on multiple sequence alignments of pomegranate AQPs with heterologous AQPs of Arabidopsis (Kumar, Stecher & Tamura, 2016). TBtools software was used to construct a chromosome localization map of AQP family genes.

Gene duplication and synteny analysis of the PgrAQP gene family

Gene duplication events of PgrAQP genes were analyzed using TBtools with the following parameters: Blast Compare Two Seqs, Outfml: Table, NumofThreads: 2, E-value:1e−5, NumofHits: 5, NumofAligns: 5; File Merger For MCScanX, Merge Mode: GtfGff2 SimGxf (Chen et al., 2020). Duplication types were divided into whole-genome (WGD), segmental, and tandem duplications (Qiao et al., 2015). Tandem duplicated genes were defined as two homologous genes within a range of 100-kb and sequence alignment coverage over 75% (Gu et al., 2002; Wang et al., 2010; Yang et al., 2008). The nonsynonymous substitution ratios (Ka), synonymous substitution ratios (Ks), and Ka/Ks ratios of the PgrAQP family genes were calculated using the TBtools simple Ka/Ks calculator program, with the following parameters: Simple Ka/Ks Calculator (NG). The Ks value was used to calculate the divergence time of duplication events (T =Ks/2R Mya, Millions of years), where R is the rate of synonymous substitutions, R = 1. 5 × 10−8 substitutions per synonymous site per year for dicotyledonous plants and T refers to divergence time (Edlund, Swanson & Preuss, 2004). The microsyntenic relationship of AQP genes between pomegranate, grape, Arabidopsis, and eucalyptus was determined using TBtools Dual Systeny Plot for MCscanX program (Chen et al., 2020).

Plant materials and analysis of transcript profiles of PgrAQP genes

Two major pomegranate cultivars, ‘Dabenzi’ (a hard-seeded cultivar) and ‘Tunisia’ (a soft-seeded cultivar), were planted in Anhui Province (Hefei, 31°51′9.05″N, 117°06′34.33″E), China and grown under the same fertilization and irrigation conditions. Seeds from five fruits of the pomegranate cultivar ‘Dabenzi’ and ‘Tunisia’ were collected at 50, 95, and 140 days after pollination (DAP). For outer seed coats, the seeds from fruits collected 95 DAP and 140 DAP was used because it was difficult to visually distinguish the inner and outer seed coats of seeds were collected at 50 DAP. Three biological replicates were taken for RNA sequencing. For each treatment, the fresh weights of the total inner and outer seed coats were recorded and immediately frozen in liquid nitrogen. Part of the inner and outer seed coats were stored at −80 °C until they were used for transcriptome sequencing. RNA extraction was performed as described previously (Qin et al., 2020). Briefly, the total RNA was isolated using a Plant RNeasy Mini Elute Cleanup kit (Qiagen, Beijing, China) and the sequencing libraries were prepared using the NEBNext Ultra™ RNA Library Prep Kit for Illumina (New England Biolabs, Ipswich, MA, USA) following the manufacturer’s protocols. Transcriptome sequencing was conducted using an Illumina HiSeq 2000 platform.

The remaining samples were used for the measurement of water accumulation. The water accumulation (%) in the different seed coats was determined by the fresh weight (FW) and drought weight (DW) using the formula (FW−DW)/FW ×100. The freeze-drying analysis was performed using GOLD-SIM FD83 (SIM International group co. ltd, USA). The experiments were repeated three times.

The abundances of pomegranate AQP transcripts, in the root, flower, leaf, and three developmental stages of the peel and seed coat (inner and outer seed coat), were collected from the microarray data previously published by Qin et al. (2017). Transcriptional abundances of AQP genes were estimated using the fragments per kilobase of exon per million mapped reads (FRKM) method, and a heat map was generated based on the log2 FRKM transformation value using TBtools (Hu et al., 2018), with the following protocol: starting from the “Graphics”, click “Heatmap lllustrator”, select the Heatmap button, then set input files in each filed, click “Start” and graph will be generated.

Verification of PgrAQP genes function responding to water deficit in pomegranate root

For the analysis of PgrAQP genes expression patterns in response to water deficit, the PEG treatment experiment was performed. For the hydroponic culture experiment, the full Hoagland nutrient solution contained 20% (v/v) polyethylene glycol 6000 (PEG6000). The solution pH was adjusted to 5.5. The roots samples were collected after 0 h, 1 h, 6 h, 12 h and 24 h of treatment, and were quickly frozen in liquid nitrogen and stored at -80 for subsequent RNA isolation.

For performing qRT-PCR analysis, approximately 2 ug of NDA-eliminated total RNA from pomegranate roots were used to synthesize cDNA using a reverse transcription kit (TaKaRa). qRT-PCR was performed on the Applied Biosystems (ABI) StepOne Plus PCR system (Applied Biosystems) using the LightCylcer 96 SYB GREEN I Master (Roche, Indianapolis, IN, USA) in a 20 ul reaction solution. The PgrAQP genes that showed higher expressed in the roots from the RNA-Seq data were selected for qRT-PCR analysis. The relative transcript abundance of each gene was normalized to the pomegranate actin (OWM91407) with the cycle threshold (Ct) 2(−ΔΔCt) method. Three biological replicates and technical replicates were used for each gene. All the specific primers used for each target gene are listed in Table S1.

Statistical analysis

The data were analyzed by ANOVA (SPSS 16.0; SPSS Inc., Chicago, IL, USA), followed by Turkey’s test (P < 0.05) to determine differences of inner and outer seed coats. The data represent the mean ± SE of three independent biological replicates.

Results

Genome-wide identification of PgrAQP genes in pomegranate reveal each Aquaporin subfamily presents particular physicochemical characteristics

A total of 38 PgrAQP members were identified in the pomegranate genome via a genome-wide search using the AQP protein sequences in Arabidopsis and Populus trichocarpa as queries (Quigley et al., 2002). A subsequent conserved domain analysis also confirmed all of the predicted AQPs (Fig. S1). The characteristics of PgrAQP family genes are shown in Table S2, including the gene ID, protein length, relative molecular weight, transmembrane domains (TMDs), subcellular localization, and calculated isoelectric point (pI). The 38 predicted PgrAQP genes encoded proteins that varied in length from 245 to 359 amino acid residues, with a relative molecular weight of 22.86 to 35.18 kDa, and a calculated pI ranging from 5.06 to 10.16. The average PI value of PgrTIPs was less than other AQPs due to the loss of basic residues in the C-terminal domain. The grand average of hydrophobicity index (GRAVY) was used to evaluated protein hydrophobicity and hydrophilicity. The results showed the GRAVY of PgrAQP proteins were all positive, ranging from 0.203 to 0.967, which indicated that all of PgrAQP proteins were hydrophobic. Furthermore, the lowest average of GRAVY value (0.44) was found in the PgrPIP subfamily, suggesting that this subfamily has better interaction with water molecules.

The predicted transmembrane domains (TMDs) showed that most PgrAQP genes (28 of 38, 73.7%) contained six TMDs, 2.7%, 15.8%, and 7.8% of PgrAQP genes contained four, five, and seven TMDs, respectively (Fig. S2). Based on subcellular localization predicted by WoLF PSORT, most PgrPIP and PgrNIP proteins were predicted to localize in plasma membranes, while only PgrPIP2.2 was found in the chloroplast. All PgrTIP proteins were predicted to localized to vacuoles. For PgrSIP proteins, PgrSIP1.1 and PgrSIP1.2 were found in the chloroplast and vacuole, respectively. PgrXIP proteins were predicted to localized in plasma membranes.

Phylogenetic characterization of the pomegranate PgrAQP gene family

To investigate the evolutionary relationship of pomegranate AQP family genes, a total of 193 AQP protein sequences from the four species studied (38 in pomegranate, 35 in Arabidopsis, 33 in grape, 55 in Populus trichocarpa and 40 in eucalyptus) were identified. An unrooted phylogenetic tree was constructed based on the alignments of their amino acid sequences in MEGA 7 using the neighbor-joining method (Kumar, Stecher & Tamura, 2016). By comparing amino acid sequences of PgrAQPs with APQs from three other plant species, 38 PgrAQPs were divided into five different subfamilies, that is, 14 PgrTIPs, 13 PgrPIPs, eight PgrNIPs, two PgrSIPs, and one PgrXIP (Fig. 1). The PgrPIPs divided into two major subgroups, PgrPIP1s and PgrPIP2s, which comprised five and eight members, respectively. Furthermore, PgrNIPs formed six subgroups (PgrNIP1, PgrNIP1, PgrNIP4, PgrNIP5, PgrNIP7, and PgrNIP8) in pomegranate. As the largest sub-family, the TIPs members were classified into five subgroups, consisting of seven PgTIP1s, two PgTIP2s, two PgTIP3s, one PgTIP4, and two PgTIP5s. SIPs and XIPs formed one group, containing two and one members, respectively. In the phylogenetic tree, PgrAQP genes were more closely related to AQPs in eucalyptus than that in Arabidopsis and grape, which is in accordance with the evolutionary relationships among these species (Qin et al., 2017).

Gene structure and conserved motif analysis of PgrAQP genes confirm the phylogenetic classification

Gene structure and conserved motifs can provide information for exploring the evolutionary relationships among a gene family. The number of introns of PgrAQP genes ranged from zero to four, while the length of exons was highly similar for each subfamily (Fig. S3). Among them, most of the PgrTIPs genes had two introns, except for PgrTIP1.6 and PgrTIP1.8, which had only one intron. For PgrPIP genes, most of members had three introns, whereas PgPIP2.8 contained two introns. The numbers of introns in PgrNIPs ranged from one to four introns. Four out of 10 members had four introns (PgrNIP2.1, PgrNIP4.1, PgrNIP4.2, and PgrNIP3.1), three members had three introns (PgrNIP1.2, PgrNIP5.1, and PgrNIP7.1), and two members had two introns (PgrNIP1.1 and PgrNIP1.3). PgrXIP2.1 had three introns. The SIP family formed a small subfamily, among which, two members had two introns and one member had no introns.

By analyzing the intron–exon structure of PgrAQP genes, we found that the number of introns of each AQPs subfamily (PIP, NIP, TIP, SIP, and XIP) are highly conserved and similar when compared with plant species, such as banana, watermelon, chickpea, and sweet orange, suggesting similar intron loss or gain events were experienced in AQP subfamily over the course of evolution (Deokar & Tar’an, 2016; Hu et al., 2015b; MartinsCde et al., 2015; Zhou et al., 2019). Multiple sequence alignments showed that PgrNIPs and PgrPIPs were the most diverse (38.5%) and conserved (72.7%) subfamily at the amino acid level, respectively (Table S3). This finding is consistent with the AQPs from Nicotiana tabacum, suggesting that the function and regulatory mechanism of PgrPIP subfamily genes showed more conserved than PgrNIP subfamily genes in pomegranate (De Rosa et al., 2020; Deokar & Tar’an, 2016). Furthermore, the relative conservation of protein sequences and ar/R selectivity filter among the PgrPIP and PgrTIP subfamily genes suggested that these proteins may share a conserved function in transporting water and other small neutral solutes (Tables S3 and S4) (Zhu et al., 2019).

To detect the structural diversity and provide further support of the grouping of PgrAQPs, a total of 10 conserved motifs were identified (Fig. S3). Generally, motif compositions were conserved within each subfamily. For PgrPIP, eight motifs were found in all family members. Motifs 1, 2, 3, 6, 8, and 10 were common for TIP and NIP subfamily members, except for PgrNIP7.1. Interestingly, motifs 4 and 5 were only identified in the PIP subfamily and similar motifs were found in bread wheat, indicating the PIP subfamily may have unique functions (Madrid-Espinoza et al., 2018).

Comparison of substrate-specific residues in PgrAQP proteins

The NPA motifs, ar/R selectively filter, and Froger’s positions were identified by multiple sequence alignment between the PgrAQPs and AtAQPs using MEGA 7.0. These highly conserved motifs and positions were critical for the substrate selectively of AQPs (Tornroth-Horsefield et al., 2006). Conserved domain searches using CDD tool from NCBI confirmed all the predicted AQP genes in Pomegranate encoded MIP domains (Fig. S3). As shown in Table S4, all of PgrTIPs, PIPs, and XIP proteins harbored two conserved NPA domains in both loop B (LB) and loop E (LE). For the PgrNIPs subfamily, except for PgrNIP5.1, the rest of numbers showed the third residue of the first NPA motifs was serine rather than alanine. In addition, PgrSIP1.2 showed the substitution of alanine by threonine in their first NPA motif.

The residues of the ar/R selectivity filter and Froger’s position displayed conserved regions within each subfamily, but regions were more variable across different subfamilies (Kayum et al., 2017). For example, all of the members of the PgrPIPs showed conservative residues at the ar/R selectively filter with phenylalanine-histidine-threonine-arginine, which is typical of aquaporin protein structure. The residues of Froger’s position were conserved in PgrPIPs, including S at P2, A at P3, Y at P4, and W at P5, while the P1 position was variable with the Q/M residues. Different residues were observed at the ar/R selectively filter and Froger’s position in PgrNIPs, which had W/G/A/V-V/S/I-A/G-R and F/L/Y-S/T-A-Y/F-I/V/LM residue compositions, respectively. For PgrTIPs members, the residues of P3, P4, and P5 positions were highly conserved, while the P1 and P2 positions were variable residues. The ar/R selectivity filters were variable with H/S/N-I/V-A/P-V/R/C residues. The residues of Froger’s position were conserved in PgrSIP1.1 and PgrSIP1.2, but the ar/R filter showed distinct difference.

The pore diameter and hydrophobicity of AQP proteins determines their substrate specificity (Almasalmeh et al., 2014; Hove & Bhave, 2011). The highly conserved amino acid features of AQPs included six transmembrane domains, the NPA domain, and the ar/R selectivity filter (Froger et al., 1998). For example, all of the PgrPIP subfamily members showed a highly conserved ar/R filter structure (T-H-R-T) (Table S4 ), which was observed in PIP family genes from other plant species, such as watermelon, Arabidopsis, Brassica rapa, soybean, and chickpea, indicating that the substrate specificity of this subfamily may be more specific than others subfamilies (Deokar & Tar’an, 2016; Kayum et al., 2017; Quigley et al., 2002; Zhang et al., 2013). Among the different TIP subgroups, the highly conserved NPA motif, ar/R H2, H5, and Forger’s P3 to P5 were observed. Furthermore, the highly conserved ar/R filter (H-I-A-V) and Froger’s positions (T-S-A-Y-W) of the PgrTIP1 subfamily were reported to function as urea and H2O2 transporter (Hove & Bhave, 2011). In addition, the conserved ar/R filter G-S-G-R residues were found in PgrNIP2.1, and this characteristic was identified as the indicator of Si transporters, indicating that PgrNIP2.1 may be involved in the transport of Si (Deshmukh et al., 2013; Deshmukh et al., 2015; Zhou et al., 2019). In pomegranate, valine was present at position H5 in the ar/R selectivity filters of PgrXIP2.1, suggesting that the hydrophobicity of PgrXIP2.1 is greater than other PgrAQPs subfamily members (Danielson & Johanson, 2008; Gupta & Sankararamakrishnan, 2009).

Segmental duplication events have contributed to the expansion of the PgrAQP family genes

To investigate the localization of PgrAQP genes and duplication events in pomegranate, we anchored the PgrAQPs on chromosomes and conducted a duplication analysis. The physical position of the PgrAQP genes were found to be unevenly distributed across all pomegranate chromosomes (Fig. 2). The PIP subfamily genes were randomly anchored on chromosomes, except for Chromosome 7, and TIP subfamily genes were found in all chromosomes except Chromosome 4. Aside from Chromosomes 2 and 8, NIP group genes were located in each chromosome. Genes in the SIP subfamily were present only on Chromosome 4.

Figure 2 Analysis of chromosomal locations and syntenic relationships of PgrAQP genes.

The AQP genes in pomegranate were mapped to different chromosomes using TBtools, and AQP genes in red and blue represent genes with segmental and tandem duplications, respectively.

We further analyzed the gene duplication modes of PgrAQP genes in pomegranate. As shown in Fig. 2, 47% of the PgrAQP genes had been duplicated by tandem/segmental duplication events. We found one tandem duplication event in Chromosome 3 (PgrTIP1.2/PgrTIP1.7). There were eight pairs of segmental duplications detected among six chromosomes. As shown in Table 1, the proportion of segmental PgrAQP gene duplications was 88%, indicating that segmental duplication events have played a key role in the expansion of the PgrAQP gene family. To access the selection pressure and the date at which such duplication events occurred, estimation of the Ka and Ks substitution rates of these duplication PgrAQP gene pairs were calculated. A Ka/Ks ratio >1 and <1 indicate positive Darwinian selection or purifying selection, whereas a value of 1 indicates neutral selection. The Ka/Ks ratios of PgrAQ P duplication gene pairs showed a Ka/Ks ratio of <1, indicating that these PgrAQP genes have experienced purifying selection during the course of evolution. According to the mathematical formula (T =Ks/2 λ) used to calculate the evolutionary date, we assessed the divergence time of these duplication events and found that the gene duplication events occurred approximately 1.74–6.97 million years ago.

Table 1 The Ka and Ks values of duplicated PgrAQP gene pairs.

Duplicated gene pairs	Duplicate type	Ka	Ks	Ka/Ks	Time (Mya)	Purify selection	
PgrTIP1.2 vs PgrTIP1.5	Segmental	0.074	1.54	0.048	5.12	Yes	
PgrTIP3.1 vs PgrTIP3.2	Segmental	0.254	1.19	0.213	3.97	Yes	
PgrPIP1.5 vs PgrPIP1.1	Segmental	0.078	0.86	0.090	2.88	Yes	
PgrPIP2.5 vs PgrPIP2.2	Segmental	0.123	1.26	0.098	4.21	Yes	
PgrPIP2.7 vs PgrPIP2.3	Segmental	0.102	1.38	0.074	4.59	Yes	
PgrPIP2.6 vs PgrPIP2.7	Segmental	0.073	1.30	0.057	4.32	Yes	
PgrTIP1.2 vs PgrTIP1.1	Segmental	0.096	2.09	0.046	6.97	Yes	
PgrTIP1.2 vs PgrTIP1.7	Tandem	0.101	0.52	0.194	1.74	Yes	
Notes.

Ka non-synonymous substitution rate

Ks synonymous substitution rate

Mya Million years ago

Further, we constructed the comparative synteny maps of three plants species (pomegranate vs. Arabidopsis, pomegranate vs. grape, and pomegranate vs. Eucalyptus) to explore the evolutionary process of PgrAQP genes (Fig. 3, Table S5) and found that 24, 28, and 41 orthologous AQP gene pairs were identified, respectively. Remarkably, the numbers of orthologous TIP gene pairs between pomegranate and grape/Eucalyptus were significantly higher than that in Arabidopsis. Nevertheless, the PIP genes were highly conserved within the species. The syntenic relationship detected in TIP genes indicates that the expansion of PgrTIPs and VvTIPs/EucTIPs genes may have occurred after that of Arabidopsis, while the PIP genes have been evolutionarily conserved.

Figure 3 Synteny analysis of PgrAQP genes between pomegranate and three plant species.

(A) Pomegranate and A. thaliana, (B) pomegranate and Vitis vinifera, (C) pomegranate and Eucalyptus grandis. The gray lines indicated collinearity between pomegranate and other species. The red lines highlight the syntenic AQP gene pairs. The chromosome name is indicated at the top of every chromosome.

Analysis of PgrAQP gene expression profiles of different pomegranate tissues and water deficit

Identifying tissue-specific genes is a basic strategy to select candidate genes involved in biological processes. To explore the possible functions of PgrAQP genes in various developmental stages of different organs of pomegranate, a heat map of Pgr AQP expression profiles was conducted. The heat map showed various expression patterns of the 38 PgrAQP genes analyzed (Fig. 4A, Table S6). Most of TIPs, such as PgrTIP1.1, PgrTIP1.4, PgrTIP1.6, PgrTIP1.8, and PgrTIP2.1, showed higher expression in the roots, leaves, and flowers, whereas the transcripts of PgrTIP2.3, PgrTIP3.1, and PgrTIP3.2 showed extremely low expression levels. For the NIPs subfamily, PgrNIP4.1 and PgrNIP5.1 were highly expressed in the leaves and roots, respectively. Interestingly, the transcripts of PgrNIP1.3 could only be detected in flowers, indicating that PgrNIP1 might be involved in the development of pomegranate flowers, while other members were transcribed at extremely low levels. The accumulation of transcripts of two PgrSIPs was detected in all tissues analyzed, whereas PgrSIP1.2 had higher relative expression levels than that of PgrSIP1.1. In the PIPs subfamily, PgrPIP3, PgrPIP2.3, and PgrPIP2.4 had a low level of transcripts in all tissues and stages analyzed, whereas the remaining members had higher expression levels in all tissues analyzed. Notably, PgrPIP1.3, PgrPIP1.5, PgrPIP2.1, and PgrPIP2.8 had higher expression levels in the peel and seed coats at all three experimental stages, indicating that these genes may play important roles in seed development.

Figure 4 Expression analysis of the PgrAQP genes in pomegranate.

(A) Expression profile of PgrAQP genes in the cultivated pomegranate cultivar ‘Dabenzi,’ including roots, flowers, leaves, and three stages of the peel, inner, and outer seed coats (50, 95, and 140 days after pollination). (B) Expression profiling of PgrAQP genes at different developmental stages of the seed coats in pomegranate. The abbreviations are as follows: D: P. granatum ‘Dabenzi,’ T: P. granatum ‘Tunisia,’ O: Outer seed coat, I: Inner seed coat. The number represents the number of days after pollination (DAP). The heat map was generated using TBtools. Expression data were calculated with log2 normalization based on FPKM values.

To reveal the potential function of PgrAQPs in response to water deficit in pomegranate root, the transcript levels of seven selected PgrAQPs from PIP subfamily for seedling under 20% PEG supply condition were tested using qRT-PCR. According to the qRT-PCR results, except the PgrPIP1.5, of which the transcript was enhanced in the roots after the 24 h of 20% PEG supply conditions, the expression of the other five paralogues, PgrPIP1.1, PgrPIP1.2, PgrPIP1.3, PgrPIP2.1 and PgrPIP1.4, were significantly repressed under the 20% PEG supply condition (Fig. 5). It should be noted that the expression levels of the PgrPIP1.2 was strongly and rapidly decreased under the 20% PEG treatment. In contrast, the expression pattern of PgrPIP2.4 showed barely changed until 12 h after 20% PEG treatment, the down-regulation was observed at 24 h.

Figure 5 Expression profiles of six Pomegranate PIPs roots in response to water deficit.

qRT-PCR was performed to determined the relative transcript level for the six PgrPIP genes. Relative expression level was normalized relative to untreated control group (0 h PEG treatment). Error bar represents SE of three independent biological replicated. Asterisk indicate significant differences, ∗P < 0.05.

Identification of candidate PgrAQPs involved in water accumulation in the outer seed coat of pomegranate

To understand water transport and accumulation in pomegranate seed coats, especially of the juicy outer seed coat, we detected water accumulation in the inner and outer seed coats at different developmental stages, as well as the relative transcript levels of PgrAQP genes in corresponding samples. We found that the water content was significantly increased in outer seed coats during seed development in ‘Dabenzi’ and ‘Tunisia,’ and the water that accumulated in the outer seed coats was higher than that in the inner seed coats (Fig. 6). Genes, including PgrPIP1.3, PgrPIP2.8, PgrPIP1.5, PgrPIP2.6, PgrPIP2.1, PgrPIP2.2, PgrPIP2.5, and PgSIP1.2, had relatively high levels of transcript accumulation in inner and outer seed coats. Among the abovementioned genes, PgrPIP1.3, PgrPIP2.8, and PgSIP1.2, had high levels of accumulation of transcripts only in the outer seed coat at the later developmental stages (Fig. 5), which indicated that these genes may be involved in the accumulation of water in outer seed coats at the later developmental stages.

Figure 6 Water accumulation in seed coats of pomegranate.

The abbreviations in the sample designations represent the cultivar names: D forP. granatum ‘Dabenzi’ and T forP. granatum ‘Tunisia.’ The second letter represents tissue: O and I stands for the outer seed coats and the inner seed coat, respectively. The numbers represent the three stages of seed coat development at different days after flowering (DAF). Error bar represents SE of three independent biological replicated. Difference letters indicate a significant difference (P < 0.05).

In Arabidopsis, a plasma membrane aquaporin, AtPIP1;2 is involved in aquaporin-mediated leaf water transport, lateral root formation, and water uptake in root. To further understand the function of PgrPIP1.3, PgrPIP2.8, and PgrSIP1.2 in the water accumulation in the seed coat, the linear relationship between the water content in the seed coat and differences in transcript levels for PrgPIP1.3 and PgrSIP1.2 in seed coat development was determined. Significant positive correlations in both PgrPIP1.3 and PgrSIP1.2 and water content in outer seed coats were observed (Fig. 7). Therefore, it was concluded that PgrPIP1.3 and PgrSIP1.2 might be involved in the accumulation of water in pomegranate seed coats.

Figure 7 Linear regression between gene expression of PgrAQPs and water accumulation.

Correlation analysis between gene expression of PgrPIP1.3, PgrPIP2.8 and PgrSIP1.2 and water accumulation in seed coats from P. granatum ‘Dabenzi’ (A, C, E) and P. granatum ‘Tunisia’ (B, D, F).

Discussion

PgrAQP identification and structure analysis

AQP proteins play an important role in various physiological and developmental processes of different organs and tissues, and function as a transporter for water and/or small neutral solutes in plants. Numerous studies have been conducted on AQPs in plants, such as in Arabidopsis, grape, soybean, rice, and Populus trichocarpa (Fouquet et al., 2008; Gupta & Sankararamakrishnan, 2009; Quigley et al., 2002; Sakurai et al., 2005; Zhang et al., 2013). However, the genome-wide identification of the AQP gene family in pomegranate was absent due to the limitations of an available genome sequence (Luo et al., 2020; Qin et al., 2017). In this study, 38 PgrAQP genes were identified and characterized in pomegranate. The characteristics of PgrAQP were comparable with other plants species. For instance, the numbers of exons and structures of intron/exons in the PgrAQP genes were highly conserved in different species, such as Arabidopsis, olive, chickpea, Arachis hypogea, and banana (Deokar & Tar’an, 2016; Faize et al., 2020; Hu et al., 2015b; Quigley et al., 2002; Shivaraj et al., 2019). Similar to the organizations of intron/exons, PgrAQP proteins of each subfamily possess the same conserved motifs (Fig. 2). These results suggest that the gene structures of PgrAQPs are closely related to homologous genes.

Furthermore, the TMDs showed that some of the PgrAQPs were lack one (PgrSIP1.1) or two (PgrNIP5.1, PgrPIP2.8) TMDs (Fig. S3). Variation in the number of TMDs to different plant species has been reported (Ayadi et al., 2011; Zhu et al., 2019). For instance, a truncated form of wheat TdPIP2;1 aquaporin, showed no water transport activity. Interestingly, the truncated tdpip2;1 could reach the plasma membrane by interact with the functional TdPIP2;1, and then may affect the functional form and reduce the water transport activity of aquaporin (Ayadi et al., 2011). Therefore, the absence of TMDs may affect the PgrAQPs subcellular localization and water transport activity.

The evolutionary relationships among the PgrAQP gene family

Gene duplication is considered as a major driving force for the evolution of gene families, and several duplication events have been identified over the course of evolution of some plant species. At approximately 117 million years ago (Mya), all core eudicots experienced a genome triplication event (the γ event), including Arabidopsis, Eucalyptus, grape, and pomegranate (Jiao et al., 2012; Jiao et al., 2014). Then, Arabidopsis experienced two recent WGDs (α and β), whereas pomegranate and Eucalyptus underwent a Myrtales lineage-specific WGD event (109.9Mya, M), but grape did not undergo any additional WGDs (Myburg et al., 2014; Qin et al., 2017). In Arabidopsis, a total of 35 AQP genes were identified and further evolution analysis revealed that AtAQPs result from different types of gene duplication, such as γ WGD (1), β WGD (2), α WGD (8), tandem (2), and transposed (4) (Bowers et al., 2003; Zwiazek et al., 2017). For poplar, tandem duplication (4) and the recent WGD (20) were the major driving forces for 55 AQP genes (Zou & Yang, 2019).

In this study, the 38-member PgrAQP family is comparable to Arabidopsis (33) and Eucalyptus (40), but had a greater number of genes than grape (28). Furthermore, all the PgrNIP, PgrSIP, and PgrXIP subfamilies were found to have a close relationship with the corresponding genes in Eucalyptus AQPs, which is consistent with the evolutionary relationships among the species, suggesting that these AQP subfamilies might have functional conservation in Myrtales (Qin et al., 2017). It is reasonable to deduce that WGD events may function as a trigger of PgrAQP family genes expansion. In additional, we identified eight gene duplication events in the PgrAQPs subfamily, including seven segmental duplications and one tandem duplication event (Fig. 2, Table 1). Interestingly, the Ka/Ks ratio of the eight duplications was <1, indicating that the evolution of the PgrAQP genes is mediated by large-scale purifying selection, similar to the AQP family in B. rapa and wheat (Kayum et al., 2017; Madrid-Espinoza et al., 2018). The synonymous substitution rate was also used to estimate the evolutionary timescale, and the divergence time of duplicated PgrAQP genes occurred 1.77 to 6.97 million years ago, which is in accordance with the divergence time of BrAQPs. This indicates that duplicated divergence of the PgrAQP genes occurred after the triplication events and Myrtales WGD duplication events (Kayum et al., 2017; Qin et al., 2017).

According to a genome-wide analysis of different organisms, it was hypothesized that the frequency of gene duplication events was important to the evolution of a species (Flagel & Wendel, 2009; Lynch & Conery, 2000). In cotton and sesame, tandem duplicated genes showed functional differentiation, although they shared structural conservation (Li et al., 2019; Wu et al, 2016). In our study, PgrTIP1.2 and PgrTIP1.7 were identified as tandem duplicated genes, and they were predominantly expressed in roots and outer seed coats in pomegranate, respectively. Our results provided more information for understanding the evolution of plant AQPs.

Potential functions of PgrAQP genes

As the largest AQP subfamily in most plant species, PIPs play a crucial role in water absorption of roots and leaves. Furthermore, PIPs can affect photosynthesis by enhancing the diffusion of CO2 in mesophyll tissue of rice and N. tabacum (Flexas et al., 2006; Xu et al., 2019). In this study, PgrPIP subfamily genes had higher expression levels in all analyzed samples compared with other analyzed AQPs. Interestingly, among the RhPIPs, RhPIP1s and RhPIP2s were involved in the expansion of rose petals via an ethylene-dependent pathway (Chen et al., 2013; Ma et al., 2008). The expression analysis showed that PgrPIP1.3 and PgrPIP2.1 had similar expression patterns in pomegranate flowers, suggesting a similar role of PIPs during pomegranate flower development (Fig. 4). In the higher plants, TIPs are widely used as markers for vacuolar compartments and function as transporter for small solutes in various tissue (Bienert et al., 2007; Holm et al., 2005; Liu et al., 2003; Porcel et al., 2018). In Arabidopsis, at least six TIP subfamily numbers, including TIP1;1, TIP1;2, TIP2;1, TIP2;2, TIP2;3, and TIP4;1, showed specific expression patterns in roots (Gattolin, Sorieul & Frigerio, 2011). In addition, the rice OsTIP2;1 was only detected in roots (Nguyen, Moon & Jung, 2013). In pomegranate, PgrTIP1.2 and PgrTIP2.3 show higher expression patterns in the roots than other organs, indicated that these gene may specifically participate in the absorption and transport of small solutes, such as NH4 +, H2O2, and urea, in the roots of pomegranate (Bienert et al., 2007; Holm et al., 2005; Liu et al., 2003). In our study, the XIP and majority of NIPs showed lower expression levels than PIPs and TIPs. Interestingly, we found that PgrNIP5.1 was highly expressed in roots. Such specific root expression of NIPs was reported for AtNIP5;1 and HvNIP2;1 (Schnurbusch et al., 2010; Takano et al., 2006), which were involved in boron (B) homeostasis, indicating that PgNIP5;1 may participate in B absorption and translocation in pomegranate roots.

Drought causes tissue dehydration due to an imbalance between plant water uptake and transpiration. Evidence shows that AQPs play an important role in drought tolerance in plants. Ectopically expression of MpPIP2;1 in Arabidopsis has been shown to enhance drought and salinity tolerance. The decrease or increase of the AQPs transcript levels could prevent water losses or helps plants to direct water flow to specific organ under drought stress. In this study, we found most members of the PgrPIP genes were suppressed by drought stress (Fig. 5), including PgrPIP1.1, PgrPIP1.2, PgrPIP1.3, PgrPIP2.1 and PgrPIP2.4, suggested their involvement in reduced water losses in Pomegranate plants. Furthermore, the upregulation of the transcript levels of PgrPIP1.5 was observed under 20% PEG supply condition, indicated that PgrPIP1.5 might play a critical for tolerance to drought in pomegranate.

Identification of candidate PgrAQP genes involved in pomegranate seed coat development

Plant cell expansion is primarily driven by turgor and requires steady water intake, the rate of tissue growth is primarily restricted due to decreasing of turgor (Peret et al., 2012; Picaud et al., 2003; Reuscher et al., 2013). For example, in higher plants, the development of a seed coat was primarily initiated by fertilization and driven by cell expansion and growth (Figueiredo et al., 2016). Accordingly, most of the AQPs were reported to be strongly expressed in tissues that can be hydraulically limited during growth. In Arabidopsis, the specific expression pattern and regulatory mechanism showed that AtTIP1;1 plays a critical role in cell expansion (Beebo et al., 2009; Ludevid et al., 1992). Overexpression of ginseng TIP in Arabidopsis resulted in a significant increase of leaf cell sizes compared with the wild type plants (Lin et al., 2007). In rice, OsPIP1;1 is highly expressed in leaves and roots, and overexpression of OsPIP1;1 exhibited a higher germination rate than the control plants (Liu et al., 2013).

In pomegranate, the expanded outer seed coats had higher water accumulation than the rigid inner seed coats in both the hard-seeded cultivar ‘Dabenzi’ and the soft-seeded cultivar ‘Tunisia’ (Fig. 6). Accordingly, significant positive correlations were found between the expression level of PgrSIP1.2, PgrPIP1.3, and PgrPIP2.8 and water content in the seed coats (Fig. 7). PgrSIP1.2 was predicted to be localized in the vacuole. Considering that promotion of cell expansion and maintenance of turgor requires the transfer of substantial amounts of water to cells, it is reasonable to proposed that PgrPIP1.3, PgrPIP2.8, and PgrSIP1.2 might be involved in mediating the water accumulation in the inner and outer seed coats of pomegranate.

Conclusions

In this study, a total of 38 AQP genes were identified and their characteristics, including protein physicochemical properties, gene structure, phyletic evolutionary, and expression patterns were studied. These PgrAQP genes are distributed across nine pomegranate chromosomes and divided into five subfamilies. Purifying selection were undergone during the evolution of PgrAQP family genes basing on the syntenic relationships and duplication events analysis, and a whole-genome duplication event in Myrtales may contribute to the expansion of PgrTIP, PgrSIP, and PgrXIP genes. Furthermore, the high expression of PgrPIP1.3, PgrPIP2.8, and PgrSIP1.2 in seed coats and the positive correlation between transcript levels of PgrAQP genes and the water content revealed these three genes may be the potential candidate genes involving in outer seed coat development. Hence, further studies on functions of this three AQP genes are needed for genetic improvement of outer seed coats in pomegranate.

Supplemental Information

Supplemental Information 1 The raw data of water content

Click here for additional data file.

Supplemental Information 2 Raw Data for PgrAQP gene RNA-Seq

Click here for additional data file.

Supplemental Information 3 Raw data for Fig. 6

Click here for additional data file.

Supplemental Information 4 PgrPIPs gene relative expression patterns

Click here for additional data file.

Supplemental Information 5 Conserved domain analysis of aquaporins identified in pomegranate using CDD tool from NCBI

Click here for additional data file.

Supplemental Information 6 Transmembrane domains analysis of PgrAQP in pomegranate

Click here for additional data file.

Supplemental Information 7 Phylogenetic relationships, gene structures and conserved motifs compositions of PgrAQPs

(A) The PgrAQPs phylogenetic tree was created by the NJ method. The five subfamilies of AQP genes were marked in orange-yellow (TIPs), blue (PIPs), pink (NIPs), red (XIP) and green (SIPs), respectively. (B) Exon-intron structure of PgrAQP genes. The boxes denote exons within coding regions, and the lines connecting them represent introns. (C) Motifs compositions of the PgrAQP proteins were identified by MEME tools.

Click here for additional data file.

Supplemental Information 8 List of primers used in this study

Click here for additional data file.

Supplemental Information 9 Characteristics of PgrAQP in pomegranate

Click here for additional data file.

Supplemental Information 10 Proteins sequence similarity matrix between members of PgrAQP genes

Click here for additional data file.

Supplemental Information 11 Conserved domains, selectively filter and amino acid residues of PgrAQP gene

Click here for additional data file.

Supplemental Information 12 Syntenic analysis of AQP in Pomegranate, Arabidopsis, grape, and eucalyptus

Click here for additional data file.

Supplemental Information 13 Expression patterns of PgrAQPs in different tissues of pomegranate plants

Click here for additional data file.

Additional Information and Declarations

Competing Interests

Author Contributions

Data Availability

The authors declare there are no competing interests.

Jianjian Liu conceived and designed the experiments, performed the experiments, analyzed the data, prepared figures and/or tables, authored or reviewed drafts of the paper, and approved the final draft.

Gaihua Qin and Jianrong Zhao conceived and designed the experiments, prepared figures and/or tables, authored or reviewed drafts of the paper, and approved the final draft.

Chunyan Liu, Jie Zhou and Bingxin Lu performed the experiments, authored or reviewed drafts of the paper, and approved the final draft.

Xiuli Liu and Jiyu Li analyzed the data, prepared figures and/or tables, and approved the final draft.

The following information was supplied regarding data availability:

The raw measurements of water content in both inner and outer seed coats, and RNA-Seq data are available in the

The raw RNA-seq data of different pomegranate tissue (SRP100581) and development stages of seed coats (PRJNA548841) are available in the Sequence Read Archive (SRA): SRX2914315.

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
