# Peer review of "Genome-wide identification of candidate aquaporins involved in water accumulation of pomegranate outer seed coat"

_PeerJ, doi:10.7717/peerj.11810_

## Round 0.1 · original submission · Major Revisions

Although this study is of potential interest for researchers in the field of aquaporin structure and function, the manuscript needs to be corrected and improved in various points. All mistakes should be taken care of, and the criticism concerning methods and data presentation need to be considered. Please respond to all the concerns risen by the referees in a point-to-point rebuttal letter.

Reviewer 1 ·

Basic reporting

English is acceptable, the literature review is not sufficient, raw data (of RNA-seq) is not shared.

Experimental design

Experimental design has flaws. See my comments below for details.

Validity of the findings

Not satisfactory. See my comments below for details.

Additional comments

The authors have identified aquaporin encoding genes in pomegranate using in silico tools and characterized the protein as well as genomic properties. Also, the transcript levels of the identified genes were studied in two different cultivars. With the data, the manuscript was constructed with organized tables and figures. However, I have several concerns over the data presented:

1. The work attempts to derive a correlation between water accumulation and expression profiles of AQP genes, but no further experiments were performed to establish this claim. Correlating water content data with the expression data of genes is not sufficiently justified with experimental proofs.

2. Line 121: While saying ‘primary gene structure,’ authors have put protein length, molecular weight, and isoelectric points within the brackets!

3. Subcellular localization was predicted using WoLF PSORT, but the data is not validated in wet-lab. Given the availability of several tools, one can do infinite ‘computational analyses’; however, validation of such data is important to derive confidence. I am also concerned about using a single tool for predicting the subcellular localization.

4. Lines 127-129 – The MSA data should be presented and studied for possible sequence variations (and their impact on the structure).

5. ‘TBtools’ has been mentioned at several places (seems, 80 % of the analyses were performed using this tool), but nowhere is the information on how it was performed, the parameters maintained, etc. are missing!

6. Section 2.3: I have concerned with studying the motifs (using MEME). Why shouldn’t the authors study the domain architecture than looking for motifs? This data should be correlated with the phylogenetic tree.

7. Gene duplication analysis is not new, and it has already been discussed in the genome sequencing paper at whole genome-level. When such comprehensive data is already available, studying the duplication of a small gene family is not scientifically significant.

8. The basis for choosing two cultivars is unclear. Also, DAPs should be justified.

9. Transcript profiling is not performed using qRT-PCR (as being done in routine genome-wide papers), but the authors have used RNA-seq. However, the information about RNA-sequencing is incomplete. How many replicate libraries are sequenced? Statistical significance of the data? How was expression measured? Data availability?

10. The raw data underlying heatmap presented to show the expression should be given in supplementary information.

11. How correct it is to correlate the RNA-seq data with microarray data of Qin et al. (2017)? Did Qin et al. paper perform the experiments in the same cultivar used in this study?

12. The ‘Results’ section lacks insights as the sub-headings summarize the data presented in tables, figures, and supplementary information.

Reviewer 2 ·

Basic reporting

Overall language is clear. I have noted some typing error or sentences which could be improved for clarity below:
• Page 8, line 69, “selectivity” spelt incorrectly.
• Page 8, lines 79-83, Sentence structure could be improved to be clearer.
• Page 8, line 85, “pollen pollination” could be changed to “flower pollination” or “pollen tube development”- unsure what you are actually referring to.
• Page 15, line 212 and 215, “plasma” spelt incorrectly.
• Page 18, line 276, “ar/R” spelt incorrectly.
• Page 28, lines 489, sentence error ‘phyletic evolutionary’, unsure of meaning.
• Page 28, line 492, typing error, ‘evolution’ spelt incorrectly.
• Page 28, line 495, ‘Furthermore’ spelt incorrectly.


Sufficient literature references and background/context provided.
• Additional references could be added to Line 73, page 8, after ‘specificity of solute transport and transport rate’.
• On Page 24, lines 401-403, the sentence is quite vague. Would be good to add a reference also.

Professional Article structure and figures displayed, all appropriate data has been made available (including raw data).

Experimental design

This research is original and seems to apply to Aims and Scopes of this journal, a research Article for Biological Science.
Although the authors mention in the introduction that aquaporins have important physiological roles within plants, there could be more specific emphasis on how this research article is could be a valuable resource for other researchers not only researching pomegranate seed coat development, but also concerned with advancing other aspects of plant performance e.g. tolerance to abiotic stresses, water and nutrient-use efficiency.

The research question is well defined. Aquaporins are vital for numerous plant physiological functions and furthering our understanding of their biology is dependant firstly on establishing the AQP gene family within a species of interest.

The investigation seems rigorous and of a good technical standard.

Methods is overall well-described. However, I have a few comments:
• The authors describe that Arabidopsis AQP sequences were used as BLAST queries to identify AQP isoforms in the pomegranate genome. It would have also been good to use AQP sequences from a species more closely related to pomegranate, such as Eucalyptus. It would allow for a more thorough search, especially as Arabidopsis does not have XIP isoforms, and pomegranate does.

• The description for methodology used for naming the pomegranate AQP isoforms is misleading. The authors state that the PIP, NIP, SIP, TIP and XIP isoforms were classified according to the nomenclature used for Arabidopsis (even though Arabidopsis does not contain XIPs). There is no further description as to how the genes we actually named considering that the phylogeny shows that the Arabidopsis AQPs do not cluster with PgrAQPs in the majority of cases (in particular for the PIPs), and as such the naming would have been assigned based on other criteria not included in the methods. The description of gene naming is often overlooked in AQP family classifications which can have implications future research interpretations i.e. mis-assigning orthologs across different species.

Validity of the findings

• In your results section, page 14-15, lines 209-210, you state that the predicted trans membrane domains (TMDs) varied from 4-7 TMDs. There is no further information as to whether the AQP isoforms containing less than 6 TMD (comprising on ~17% of the genes identified) would actually be functional genes, or whether they might be eroding/non-functional genes. AQPs have a highly conserved structure across all kingdoms of life, having 6 TMDs, so a shorter sequence or altered structure would greatly impact function and tetramer formation.
• On page 17, line 256, you mention that your finding of PgrNIPs and PgrPIPs being the most and least conserved subfamilies at amino acids level, respectively, was NOT consistent with findings in Nicotiana tabacum. This interpretation is incorrect, as your findings are actually in agreement with finding from Nicotiana tabacum.


The data on which the conclusions are based were provided.
Conclusions are well stated, linked to original research question & limited to supporting results.

Additional comments

No Additional comments.

---

## Round 0.2 · Minor Revisions

Over all, the manuscript has been improved according to the previous reviewing suggestions. However, there are still some minor points that should be incorporated, in particular the evidence and facts for the naming according to the P. trichocarpa AQP isoforms, which should be included into Fig. 1.

Also, functionality of the truncated forms should be commented on in Table 1.

Finally, Water transporting activities should be given in more detail when making the point of PIP and NIP family homology.

Please submit a point-by-point statement on all changes made in the new version.

Reviewer 1 ·

Basic reporting

No comment

Experimental design

No comment

Validity of the findings

No comment

Additional comments

No comment

Reviewer 2 ·

Basic reporting

- Grammatical/sentence error on Page 8, line 77-80. I commented on this in the previous report. I would amend to “The AQP gene family has been widely studied in numerous plant species, such as…”.
- Page 8, line 85, Species name should be provided before introducing as “PvPIPs”.
- Page 9, line 90, Species name should be provided before introducing as “PsNIPs”.

Previous comments requesting additional references have been addressed.
- Page 26, line 450, reference missing at the end of sentence: “Variation in the number of TMDs to different plant species has been reported (reference)”

Experimental design

Clarifications have been provided in regards to the identification of the XIP isoforms in pomegranate, using Populus trichocarpa XIPs.

The authors have addressed my concern relating to the naming methodology of the pomegranate AQP isoforms. In the methods, however, you still state that the “Pomegranate AQPs were named according to the standardized MIP nomenclature” which is not very informative, as there isn’t really such a thing.
Instead, the methods description would be improved by mentioning that the PgrAQps were named based on homology to Populus trichocarpa AQP isoforms, and state why you chose this species to guide your naming.
I have a question as to why you did not include the Populus trichocarpa AQP sequences in your Figure 1 Phylogeny, if they were actually the basis of your naming.

Validity of the findings

Additional information was added in the discussion in regards to the impaired functionality of truncated AQP isoforms in other species (e.g. truncated TaPIP2;1 had no water function activity). Functional AQP proteins generally have 6 TMDs, therefore a clearer indication could be added by the authors as to whether the PgrAQPs which have less than 6 TMDs are even functional (these could be noted in a column in Table 1 for example). This would be important for researchers looking to do further study on your identified proteins, as one might not want to do further study on a truncated/non-functional protein.

The sentence on page18, line 283-287: “Multiple sequence alignments showed that PgrNIPs and PgrPIPs were the most diverse (38.5%) and conserved (72.7%) subfamily at the amino acid level, respectively (Table. S3). This finding is consistent with the AQPs from Nicotiana tabacum, suggesting that PgrNIP and PgrPIP subfamily genes may have a similar role in water transport in pomegranate” is not clear. It implies that the authors are inferring water permeability based on based on within family homology/diversity of PIP and NIP sequences (although NIPs are generally poor water transporters)? More explanation/reasoning could to be included if you were to make this statement.

Additional comments

The line numbers in the ‘Rebuttal’ document do not match up with the amended document. This made following up the revisions difficult. I would strongly encourage the authors to ensure that in future manuscripts the line numbering can be clearly followed.

---

## Round 0.3 · accepted · Accept

After this round of revision, your manuscript has been definitely improved and can serve as a valuable source for further studies.